# Surface-Pore-Modified N-Doped Amorphous Carbon Nanospheres Tailored with Toluene as Anode Materials for Lithium-Ion Batteries

**DOI:** 10.3390/nano14090772

**Published:** 2024-04-28

**Authors:** Shiran Shan, Chunze Yuan, Guangsu Tan, Chao Xu, Lin Li, Guoqi Li, Jihao Zhang, Tsu-Chien Weng

**Affiliations:** 1School of Physical Science and Technology, ShanghaiTech University, Shanghai 201210, China; shanshr@shanghaitech.edu.cn (S.S.); tangs@shanghaitech.edu.cn (G.T.); xuchao1@shanghaitech.edu.cn (C.X.); lilin1@shanghaitech.edu.cn (L.L.); ligq1@shanghaitech.edu.cn (G.L.); zhangjh5@shanghaitech.edu.cn (J.Z.); 2Center for Transformative Science, ShanghaiTech University, Shanghai 201210, China

**Keywords:** lithium-ion battery, amorphous carbon nanospheres, anode, soft template

## Abstract

The surface modification of amorphous carbon nanospheres (ACNs) through templates has attracted great attention due to its great success in improving the electrochemical properties of lithium storage materials. Herein, a safe methodology with toluene as a soft template is employed to tailor the nanostructure, resulting in ACNs with tunable surface pores. Extensive characterizations through transmission electron microscopy (TEM), scanning electron microscopy (SEM), Raman spectroscopy, X-ray diffraction (XRD), X-ray photoelectron spectroscopy (XPS), and nitrogen adsorption/desorption isotherms elucidate the impact of surface pore modifications on the external structure, morphology, and surface area. Electrochemical assessments reveal the enhanced performance of the surface-pore-modified carbon nanospheres, particularly ACNs-100 synthesized with the addition of 100 μL toluene, in terms of the initial discharge capacity, rate performance, and cycling stability. The interesting phenomenon of persistent capacity increase is ascribed to lithium ion movement within the graphite-like interlayer, resulting in ACNs-100 experiencing a capacity upswing from an initial 320 mAh g^−1^ to a zenith of 655 mAh g^−1^ over a thousand cycles at a rate of 2 C. The findings in this study highlight the pivotal role of tailored nanostructure engineering in optimizing energy storage materials.

## 1. Introduction

Due to their high energy density, high coulombic efficiency, and long cycle life [1,2,3,4], rechargeable lithium-ion batteries (LIBs) are favored for the widespread use of everyday electronic devices and electric vehicles [3]. The anode material as an important component of LIBs, playing an important role in their safety, capacity, and cycle life. Graphite is a very common commercial anode material for LIB; however, its theoretical capacity is limited to only 372 mAh g^−1^ and has a poor rate capability [5]. This limitation hinders the battery’s overall performance, making it insufficient to keep up with the growing demands of the electronics market.

A great deal of effort has been devoted to the exploration of alternative anode materials with improved capacity and rate performance to replace graphite [6,7,8,9,10,11,12,13]. Due to their wide range of sources, tunable ability, strong structure, short diffusion path, etc., nanostructured carbon materials were considered to be a superior choice for future high-performance LIBs [14,15,16,17,18,19,20]. Extensive research has been conducted on a variety of nanostructured carbon materials as potential anode materials for LIBs, including graphene [13,18,21], carbon nanospheres [15], carbon nanotubes [19], carbon fibers [16], and carbon quantum dots [22]. Among them, carbon nanospheres have received a lot of attention due to their large specific surface area and ease of tuning and decorating, which can increase their contact area with the electrolyte and shorten the diffusion path for lithium ions [23]. Additionally, its robust internal structure ensures higher stability, to improve the cycle stability and rate performance of LIBs, thus providing long-term cycle use and fast charging performance in practical applications [2].

To shorten the diffusion path of mass transfer, substantial research efforts have focused on the design and manipulation of carbon nanospheres, achieved by reducing the diameter of carbon spheres to the nanometer scale, thereby enhancing the abundance of exposed surface active sites [23,24,25]. For instance, researchers have synthesized carbon spheres with a diameter of about 200 nm, yielding a specific capacity of 378 mAh g^−1^ at a rate of 0.1 C [24]. Furthermore, an alternative approach has involved the utilization of templates to regulate the surface of carbon nanospheres, resulting in a larger specific surface area, shorter transfer path, and large pore volume. For example, SiO_2_ served as a hard template for the surface modification of carbon spheres, yielding a maximum specific surface area of 1117 m^2^ g^−1^ and a pore volume of 1.77 cm^3^ g^−1^ [26]. However, the use of HF to remove SiO_2_ undoubtedly increases the safety risk of the overall synthesis procedure. In another study, mesoporous carbon nanospheres featuring surface pores showed an enhanced specific capacity of 560 mAh g^−1^ at a rate of 0.1 C, approximately twice that of unmodified carbon nanospheres, and exhibited a high capacity of 240 mAh g^−1^ at an increased rate of 5 C [27]. In addition, by introducing single or double heteroatoms (boron, nitrogen, sulfur, oxygen, and phosphorus) into the carbon skeleton, the electronic and chemical structure can be adjusted, which can improve the electronic conductivity, expand the interlayer distance, improve the absorption of Li^+^/ electrolyte, and provide more active sites to improve the storage and recycling performance of Li [28,29,30]. Nitrogen (N) atoms have a high electronegativity and atomic diameter, similar to carbon, and are the most widely used heteroatoms, being able to provide free electrons to the π system of carbon to improve its conductivity [31] and introduce active sites to improve the electrochemical reaction performance and obtain a higher lithium ion storage capacity [32]. For example, Wang et al. [29,30] demonstrated that the specific capacity can be increased to 600 mAh g^−1^ when 3.9% N is doped in a graphitic carbon framework. The calculation results showed that additional Li storage capacity (higher than 395.21 mAh g^−1^) can be generated at the nanopore and N-doped edges due to the presence of pyridine and pyrrole nitrogen as active sites for the adsorption of Li atoms.

In this study, we presented a safe and conventional synthesis route of amorphous carbon nanospheres (ACNs) and investigated the influence of their surface morphology on lithium storage when used as an anode material. For safe synthesis, toluene served as a soft template to control the surface morphology of the ACNs. Four ACN samples, each featuring distinct surface pore sizes, were prepared by adjusting the amount of toluene during synthesis. The synthesized ACNs were characterized by transmission electron microscopy (TEM), scanning electron microscopy (SEM), Raman spectroscopy, X-ray diffraction (XRD), X-ray photoelectron spectroscopy (XPS), and nitrogen adsorption/desorption isotherms. The ACNs assembled as lithium-ion battery anodes exhibited a significantly enhanced electrochemical performance when using surface-pore-modified nanospheres compared to the unmodified nanospheres. The modified nanospheres demonstrated a remarkable reversible capacity of up to ~600 mAh g^−1^ at a rate of 0.05 C. In cycling stability tests at a rate of 2 C, all the ACN samples displayed capacity increases with cycling over 1000 cycles. Notably, the best-performing sample, ACNs-100 (with 100 μL toluene added in synthesis), demonstrated an increase in capacity from an initial 320 to a peak of 655 mAh g^−1^ in 1100 cycles.

## 2. Experimental Procedures

### 2.1. Materials

Triblock poly(ethylene oxide)-b-poly(propylene oxide)-b-poly(ethylene oxide) Pluronic F-127 (PEO106PPO70PEO106, Mw = 12,600 g mol^−1^) was obtained from Sigma-Aldrich (Shanghai, China). Ammonium persulfate was purchased from Maclin (Shanghai, China). 1,3,5-Tris (4-aminophenyl) benzene (TAPB) was purchased from Adamas (Shanghai, China). Anhydrous ethanol, hydrochloric acid, and toluene were obtained from Sinopharm Chemical Reagent Co., Ltd. (Shanghai, China). All chemicals were of analytical grade and used without further purification. Deionized water was taken from a YISHUO Purification system (Shanghai, China) for all experiments.

### 2.2. Preparation of N-Doped Amorphous Carbon Nanospheres (ACNs)

N-doped ACNs were prepared by using TAPB as a nitrogen and carbon source, Pluronic F-127 as surfactant, and ethanol and water as organic cosolvent agent, followed by catalytic polymerization and carbonization. In a classic synthesis, 0.1 g Pluronic F-127 and 100 μL of toluene were dispersed in 6 mL solvent mixture of water and ethanol (1:1 in volume) and ultrasonicated for 20 min to obtain an emulsion solution at room temperature. Then, 20 mg TAPB (dissolved in 0.5 mL 0.5 M hydrochloric acid) was slowly added to the solution under vigorous stirring to form a nanoemulsion system. After stirring for 1 h, 0.1 g ammonium persulfate (4.8 wt% in water) was injected into the above mixture and stirred for 12 h continuously to induce the self-polymerization of 1,3,5-Tris(4-aminophenyl) benzene oligomers. The N-doped resin nanospheres were obtained by centrifuging and washing with tetrahydrofuran, deionized water, and ethanol in sequence at least three times. Then, the resulting products were dried at 60 °C for 24 h in an oven.

The N-doped amorphous carbon nanospheres could be easily produced after pyrolyzation of N-doped resin nanospheres by preheating at 350 °C for 3 h and further heating at 800 °C for 2 h with a heating rate of 1 °C min^−1^ under a N_2_ atmosphere.

The surface pore sizes of the carbon nanospheres could be customized by adjusting the amount of toluene in the reaction system. In this work, different volumes of toluene (10 μL, 50 μL, 100 μL, and 200 μL) were used to form various N-doped resin nanospheres with adjustable pore sizes. After the carbonization, the corresponding N-doped amorphous carbon nanospheres were obtained.

### 2.3. Characterization

TEM was carried out at 200 kV with the JEM 2100Plus (JEOL, Akishima, Japan). SEM analysis was performed using a Zeiss GeminiSEM450 microscope (Jena, Germany). XRD patterns were collected with Cu Kα radiation (40 kV, 40 mA) by using the Bruker D8 (Karlsruhe, Germany). Nitrogen adsorption/desorption isotherms at 77 K were characterized by using a Quantachrome Autosorb-iQ-MP-AG (Boynton Beach, FL, USA). XPS was collected using a Thermo Fisher ESCALAB 250Xi (Shanghai, China) with a monochrome Al source. All the binding energies were modified using the C 1 s standard peak at 284.6 eV. Raman spectra were recorded on a microscopic Raman spectrometer (ANDOR SR-500I-D2-1F1 500 mm focal length, motorized Czerny-Turner Spectrograph, UK) (TechnoSpex Pte Ltd., Singapore), using a He-Ne laser with an excitation wavelength of 532 nm.

### 2.4. Electrochemical Measurements

The slurry was prepared by mixing 70 wt% of active materials (ACNs-10, ACNs-50, ACNs-100, ACNs-200), 10 wt% of binder (polyvinylidene fluoride, PVDF) in N-Methylpyrrolidone, and 20 wt% of Super P carbon black. The working electrode was prepared by casting the slurry onto copper foil and dried at 60 °C in a vacuum oven for 12 h. The areal mass loading of active materials was ~0.25 mg cm^−2^. The coated copper foil was cut into 12 mm diameter disks. The CR 2032 coin cells were assembled in an Ar-filled glovebox. Lithium metal foil (Φ15.6 × 0.45 mm) was used as counter electrode and the Celgard 2500 polypropylene membrane was used as separator. A concentration of 1.0 M LiPF_6_ in a 3:7 (*v*/*v*) mixture of ethylene carbonate and ethyl methyl carbonate was adopted as the electrolyte. Galvanostatic discharge/charge and rate capacity were operated with a voltage range of 0.01–3 V (vs. Li+/Li) on the Neware CT-4008Tn test system (BTS Client 8.0.0.575 version) (Shenzhen, China). Cyclic voltammetry (CV) was evaluated at a scan rate of 0.1 mV s^−1^ in the voltage window between 0.01 and 3 V with a Bio-Logic VMP3 FlexP 0160 instrumentation system (Seyssine, France). Electrochemical impedance spectroscopy (EIS) was conducted in the frequency range from 0.01 Hz to 100 kHz.

## 3. Results and Discussion

To enhance the safety of the synthesis processes for ACNs, we employed a safe methodology, outlined in Figure 1. Resin nanospheres with different surface pore sizes were synthesized by multi-purpose nanoemulsion method [33], using 1,3,5-tris (4-aminophenyl) benzene (TAPB) as a carbon source, Pluronic F-127 as a surfactant, and toluene as a soft template. Toluene played a key role in tuning the surface pore size, promoting enhanced hydrophobic interaction between toluene and F-127-derived polypropylene oxide segments [34]. The F-127/toluene/TAPB nanoemulsion in a mixed system of water and ethanol served as the nucleation site of TAPB polymerization during the synthesis process. As the polymerization progressed, the resin nanospheres underwent growth with F-127/toluene/TAPB nanoemulsions adsorbing onto their surface to prevent aggregation. This protective mechanism resulted in the formation of resin nanospheres with ordered porous surfaces [35]. The polymerization with (NH_4_)_2_S_2_O_8_ as a catalyst in the complex emulsions further facilitated the fusion of thermodynamically unstable emulsions [33], followed by the removal of toluene to obtain resin nanospheres with different pore sizes. In the final step, these resin nanospheres were subjected to sintering and carbonization processes to produce four distinct types of ACNs, denoted as ACNs-10, ACNs-50, ACNs-100, and ACNs-200, signifying the utilization of 10, 50, 100, and 200 μL of toluene in the synthesis process, respectively.

SEM and TEM, as depicted in Figure 1, revealed the unmodified surface morphology of the ACNs-10 with a diameter of ~400 nm. In contrast, the diameters of the ACNs-50, ACNs-100, and ACNs-200, featuring surface pore modification, fell within the range of 40~60 nm. The substantial size difference between the ACNs-10 and other samples arose from the inadequacy of 10 μL of toluene used in the synthesis of the ACNs-10 to effectively interact with F-127, preventing the formation of a stable F-127/toluene/TAPB nanoemulsion. Under this condition, toluene and F-127 collaborate to stabilize the resin nanospheres by adjusting the surface amphiphilicity, yielding smooth, large-sized, non-porous resin nanospheres [36]. With the addition of an increased amount of toluene (≥50 μL), a stable F-127/toluene/TAPB nanoemulsion forms, facilitating TAPB polymerization and producing smaller-sized resin nanospheres. The surface porous structure is favorable for electrolyte infiltration and penetration of the anode material, effectively shortening the diffusion length of the lithium ions [37]. Compared to the ACNs-50, both ACNs-100 and ACNs-200 exhibited larger and deeper surface pores, indicating the tunability of the surface pore size of the nanosphere through adjustments to the addition of toluene. In addition, when utilizing 10, 50, and 100 μL of toluene, the resulting emulsions exhibited clarity and transparency. However, with the use of 200 μL of toluene, a turbid mixed liquid was generated (Appendix A). Hence, it can be inferred that excess toluene does not contribute to the formation of the nanoemulsion. This observation may account for the similarity in the size and surface morphology between the ACNs-100 and ACNs-200. Furthermore, in Appendix A, a disordered graphitic structure within the ACNs was observed, displaying predominantly discontinuous graphite-like layers with numerous internal defects. This feature may be beneficial for lithium ion storage, as suggested by previous literature [5].

Further XRD characterization revealed the presence of two broad peaks for the ACNs (Figure 2a). Notably, the (002) peak exhibited a substantial reduction in intensity and broadening compared to the XRD pattern of the resin nanospheres before carbonization (Appendix A). This observation emphasized the limited degree of graphitization within the carbon nanospheres, indicating an inherently disordered carbon structure. This was also confirmed by the results of the Raman spectroscopy (Figure 2b), which exhibited two peaks at ~1350 and ~1590 cm^−1^, representing sp3 carbon (D band) and sp2 carbon (G band), respectively, with a calculated I_D_/I_G_ ratio of 1.25. It was clearly observed that the intensity of the D band surpasses that of the G band, indicating a significant presence of defects and disordered non-graphitic carbon structures in the nanospheres [38]. The nitrogen adsorption/desorption isotherms are shown in Figure 2c. The specific surface area (S_BET_) was calculated using the Brunauer–Emmett–Teller model, while the micropore surface area (S_MICRO_), micropore volume (V_MICRO_), and external surface area (S_EX_) were calculated using the t-plot method. The corresponding results are summarized in Appendix A. It was evident that the S_BET_ and S_MICRO_ of the ACNs-10 are significantly smaller and larger, respectively, compared to the other samples. This is primarily attributed to its larger particle size and the absence of surface pore modification. In the case of the pore-modified ACN samples, the ACNs-100 exhibited the highest S_BET_, at ~618 m^2^ g^−1^. As the surface pore size and depth increase, there is a gradual decrease in the S_MICRO_ and a corresponding increase in the S_EX_. The similar S_EX_ observed in the ACNs-100 and ACNs-200 suggested a potential similarity in surface morphology, which is consistent with the SEM and TEM findings.

The full spectrum (Figure 2d) of the XPS revealed three clear peaks, corresponding to C 1 s, N 1 s, and O 1 s in the ACNs, and, notably, there were no impurities present. The elemental contents of C, N, and O are 89.2, 5.0, and 5.8 wt%, respectively. In the XPS C 1 s spectrum (Figure 2e), three distinct carbon species, namely, C = C, C − N, and O − C = O, can be identified, each exhibiting binding energies of 284.6, 285.9, and 289.4 eV, respectively [34]. In addition, as depicted in Figure 2f, the N 1 s spectrum can be fitted into three discernible peaks: pyridinic N at 398.5 eV, pyrrolic N at 400.9 eV, and graphitic N at 403.6 eV [39,40]. The pyridinic N and pyrrolic N in the ACNs can constitute a lot of defects and serve as the active site for lithium storage, thus increasing the lithium battery capacity [5,40].

To evaluate the electrochemical performance of the ACN anode, a Li/ACN lithium-ion half battery was assembled in a button cell, employing thin lithium metal foil as the counter electrode, ACNs as the working electrode, and LiPF_6_ as the electrolyte. At a charging and discharging rate of 0.05 C, the initial discharge (lithiation) capacities of the battery with ACNs-10, ACNs-50, ACNs-100, and ACNs-200 electrodes were 698, 985, 1159, and 1123 mAh g^−1^, respectively (Figure 3a). The presence of disordered carbon and a micropore structure within the ACNs contributes to their increased lithium storage capacity and enhanced ion transport channels [24,26]. Simultaneously, surface pore modification adds to the external surface area of the ACNs, facilitating a broader contact area with the electrolyte. This exposure of more active sites serves to reduce ion transport paths, ultimately offering a greater storage capacity for lithium ions [24,27]. In the subsequent rate performance test cycled at different rates from 0.05 C to 25 C, as shown in Figure 3b, the surface-pore-modified ACNs consistently outperformed the unmodified ACNs-10. This enhancement is attributed to their significantly larger external surface area. The ACNs-100 consistently maintained the highest discharge capacity at each rate, with a capacity of approximately 600 mAh g^−1^ at a low rate of 0.05 C and 145 mAh g^−1^ at a high rate of 25 C. This is attributed to its small size and relatively large specific surface area and external surface area. This characteristic has great potential in fast charging applications. Upon returning to 0.5 C at the end of the rate performance test, the capacity of the Li/ACN batteries showed an increase. The ACNs-200 exhibited a similar capacity to the ACNs-100 at various rates, which corresponds to their morphology being almost identical to that of the ACNs-100.

The charge–discharge curve at a rate of 0.05 C and cyclic voltammetry (CV) curve at a scan rate of 0.1 mV s^−1^ of the ACNs-100 battery are shown in Figure 3c,d. In the initial cycle, the specific capacity of discharge (lithiation) and charge (delithiation) were 1159 and 566 mAh g^−1^, respectively, with a Coulombic efficiency of 49%. The considerable irreversible capacity is mainly attributed to the formation of the solid electrolyte interphase (SEI) and the presence of an irreversible active site [41,42]. Furthermore, it was observed that the voltage plateau and redox peak are not well-defined when the battery capacity has stabilized, indicating that the ACNs-100 electrode has multiple lithium storage behaviors. The capacity in the voltage range of 2.0–3.0 V is mainly attributed to the binding of the lithium ions with nitrogen atoms and physical adsorption, while the capacity in the voltage range of 0.8–2.0 V arises from the trapping of lithium ions by the defects in the ACNs-100 [37,43,44,45]. The region with a voltage below 0.8 V contributes 75% of the capacity, which is attributed to lithium ion intercalation into the graphite layers, and the voltage slope implies a disordered stacking of the graphite sheets [37,43].

Figure 4 shows the cycling stability test of the Li/ACN batteries at a rate of 2 C for 2000 cycles. Interestingly, the capacity of all the ACN batteries increased with the growing cycle number. ACNs-100 and ACNs-200 achieved their maximum capacities at the ~1100th cycle before experiencing decay, while the ACNs-50 and ACNs-10 maximized their capacities at the ~1300th and ~2000th cycles, respectively. The order of their maximum capacities is ACNs-100 > ACNs-200 > ACNs-50 > ACNs-10, signifying that the surface pore modification of the carbon nanospheres contributes to enhanced battery performance, with a larger surface pore demonstrating superior performance. In the case of the ACNs-100, the specific capacity of the battery initiated from 320 mAh g^−1^ and steadily rose, reaching a maximum of 655 mAh g^−1^ after ~1100 cycles. The capacity of the ACNs-200 closely paralleled that of the ACNs-100 throughout the entire stability cycling, aligning with their rate performances. The similar morphology, surface area, and battery performance of the ACNs-100 and ACNs-200 suggested that the maximum amount of toluene addition in the synthesis process is approximately 100 μL.

To investigate the persistent capacity increase observed over a thousand cycles, we employed electrochemical impedance spectroscopy (EIS) to gain a deeper understanding of the kinetic behavior of the ACN electrodes, both before cycling and after long-term cycling. As shown in both Nyquist diagrams before and after cycling (Figure 5), using the Li/ACNs-100 battery as an illustrative example, clear sunken semicircles were observed in the middle- and high-frequency region, indicating the presence of charge transfer resistance. The charge transfer resistance for the battery after cycling is less than 50 Ω, a substantial reduction compared to the pre-cycling resistance of ~900 Ω. This substantial reduction in charge transfer resistance after cycling is conducive to the implementation of lithium ion transport for storage, providing partial insight into the observed capacity increase in the Li/ACN battery during cycling. This phenomenon of capacity increase, previously reported multiple times [38,40,46], can be attributed to the reciprocal movement of lithium ions into and out of the graphite-like interlayer structure within the ACNs during charge and discharge cycling, resulting in an expanded spacing between the graphite-like interlayers, consequently creating more room for lithium storage [47]. Another explanation is that the amorphous carbon material broke due to the reciprocating insertion of lithium ions during the charging and discharging process, exposing more active sites and defects [48]. The XRD pattern of the ACNs-100 after cycling (Appendix A) revealed the absence of the original (002) and (100) peaks within the amorphous carbon nanospheres, indicating the complete disordering of the internal structure. It is hypothesized that the repeated lithium ion insertion and removal may disrupt the ordered structure within ACNs, potentially creating additional storage space and, therefore, increased capacity. These conditions allow for more lithium to be embedded in the amorphous carbon in the next cycle, increasing the discharge capacity.

## 4. Conclusions

In conclusion, a safe methodology was employed to synthesize ACNs with controlled surface pore modification, achieved through the precise control of toluene addition during the nanoemulsion process. SEM and TEM revealed distinct surface morphologies for various ACNs. XRD and Raman spectroscopy provided insights into their structural characteristics, highlighting the inherently disordered carbon structure of the ACNs. The nitrogen adsorption/desorption isotherms indicated a direct correlation between the external surface area and surface pore modification. The XPS exhibited elemental compositions of C, N, and O in the ACNs. The electrochemical measurements of the initial discharge capacity, rate performance, and cycling stability confirmed the enhanced performance of the surface-pore-modified ACNs, with the ACNs-100 exhibiting the most outstanding results. A persistent rise in capacity was observed in the cycling test, notably in the ACNs-100, where the specific capacity increased from an initial 320 mAh g^−1^ to a peak of 655 mAh g^−1^ at a rate of 2 C. This phenomenon was attributed to the reciprocal movement of the lithium ions within the graphite-like interlayer of the ACNs. The EIS provided evidence of reduced charge transfer resistance after cycling, contributing to enhanced lithium-ion transport and storage. The observed improvements in the electrochemical performance underscore the significance of tailored nanostructure engineering in advancing the field of energy storage materials.

## Data Availability

Data are available on demand.

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
