# Peer review of "Surface-Pore-Modified N-Doped Amorphous Carbon Nanospheres Tailored with Toluene as Anode Materials for Lithium-Ion Batteries"

_nanomaterials, 2024, doi:10.3390/nano14090772_

Round 1
Reviewer 1 Report
Comments and Suggestions for Authors
Present work is on the fabrication of amorphous carbon nanosphere and its lithium ion battery characteristics.
It is an interesting work and should be suitable to Nanomaterials.
There are several points that needs to be addressed.
i) The materials prepared in the paper is a nitrogen doped ACN. However, there is no mention on the N-doping in the title and in the introduction. Papers related to nitrogen doping in nanocarbon materials for LIB should be added as well.
ii) Please refer to Materials Today Nano Volume 22, June 2023, 100321
S. Ali, et al. Phys. Status Solidi A, 219 (8) (2021), Article 2100714,
and show how present work stands out from previous similar studies.
iii) Fig 2e and f are misplaced. It should be the other way round.
iv) In lines 245-246, the authors claim as though it was only the capacity of ACN100 that has increased at 0.5C. However, all the samples showed similar trend. The text should be improved.
v) Regarding Fig4, ACN10 becomes superior to other samples after ~2000 cycles. Isn't this a good thing for LIB? Please explain why ACN100 is still your materials of choice for LIB.
Reviewer 2 Report
Comments and Suggestions for Authors
The manuscript entitled, “Surface Pore-Modified Amorphous Carbon Nanospheres Tailored with Toluene as Anode Materials for Lithium-Ion Batteries" presents the modified carbon for energy storage applications. The research is interesting, and the authors have presented the manuscript well. I recommend a minor revision and address the following points.
1. The authors should provide the intensity ratio of ID/IG.
2. It is better to fit the EIS data as presented in Fig. 5.
3. In Fig. 4, the authors presented an increase in the specific capacity with the cycle number. The authors should explain the reason behind it. Also, it is better to provide the SEM and XRD of the materials after the cycle.
4. The authors should include some references to the current research work.
Comments on the Quality of English LanguageMinor editing of English language required
Reviewer 3 Report
Comments and Suggestions for Authors
In this manuscript, authors investigated the surface pore-modified amorphous carbon nanospheres tailored with toluene as anode materials. Although it is interesting, there are certain parts that must be addressed:
1. Please justify the reason of choosing surface pore-modified amorphous carbon nanospheres tailored with toluene as the anode material.
2. Computational supporting might be interesting to make this study more solid.
3. Compared to the state of art materials of anode, how this suggested system is attractive?
4. English typos must be carefully checked throughout the manuscript.
Comments on the Quality of English LanguageN/A
Round 2
Reviewer 3 Report
Comments and Suggestions for Authors
Authors answered to each comment I provided, hence, no more comments to authors. It can be accepted for Nanomaterials as it is.
Comments on the Quality of English LanguageN/A